# Coexistence of Thumb Aplasia and Cleft Lip and Alveolus with Aortopulmonary Window—A Tip for Prenatal Diagnostics for Rare Heart Anomalies

**DOI:** 10.3390/diagnostics12030569

**Published:** 2022-02-23

**Authors:** Anna Kasielska-Trojan, Barbara Święchowicz, Bogusław Antoszewski

**Affiliations:** Plastic, Reconstructive and Aesthetic Surgery Clinic, Medical University of Lodz, Kopcinskiego 22, 90-153 Lodz, Poland; swiechowicz.barbara@gmail.com (B.Ś.); boguslaw.antoszewski@umed.lodz.pl (B.A.)

**Keywords:** aortopulmonary window, cleft lip, thumb aplasia

## Abstract

Multiple congenital anomaly syndromes pose a challenge to neonatologists, as many anomalies may indicate cryptogenic malformations or disorders. Aortopulmonary window (APW) is a very rare congenital heart disease (CHD) and causes many difficulties in prenatal diagnostics. In this report, we describe a case of a female patient with multiple rare congenital malformations: aortopulmonary window, right thumb aplasia, facial nerve palsy and cleft lip and alveolus. None of the malformations were diagnosed prenatally. A long-term follow-up (40 years) is presented. The presence of certain defects (thumb aplasia) may indicate the need for a careful fetal examination extended by a fetal ECHO performed in a reference center of prenatal cardiology. The coexistence or syndromic character of the presented malformations should be verified in future if more such cases are described.

## 1. Introduction

Aortopulmonary window (APW) is a very rare congenital heart disease (CHD) that results in abnormal communication between the proximal aorta and the main pulmonary artery, but where the aortic and pulmonary valves are separated [1]. It is one of the rarest congenital heart diseases; the prevalence is approximately 0.1–0.3% of all cardiac malformations [1]. During embryonic life, the aortopulmonary septum forms at nine weeks of gestation [2]. APW can occur as an isolated disease or can coexist with other heart abnormalities (patent ductus arteriosus, interrupted aortic arch, coarctation of the aorta, transposition of the great arteries, tetralogy of Fallot, ventricular septal defect and atrial septal defect). Three types of APW have been distinguished: I—proximal (the most common); II—distal; III—complete. Clinical manifestations can vary from asymptomatic in the case of small defects to pulmonary hypertension and congestive heart failure symptoms. In the first months of life, pulmonary hypertension may appear and increase with age [3].

Thumb aplasia is the most severe form of thumb hypoplasia, classified as type V (complete absence of the first digit), and occurs in approximately 0.25/10,000 births. This malformation has a higher prevalence in males than in females. Primarily, it is associated with other anomalies, so isolated cases are very rare. Thumb agenesis most commonly occurs in genetic syndromes such as Holt–Oram syndrome, Fanconi anemia, VACTERL association, Seckel syndrome, thrombocytopenia-absent radius syndrome and Duane-radial ray syndrome. In most of these syndromes, the following systems and organs are also affected: renal, skeletal, hematological, heart and orbits [4].

Cleft lip and alveolus (cleft of the primary palate) is often a nonsyndromic anomaly mediated by genetic and environmental factors (smoking, alcohol consumption and taking drugs). In 70% of cases, this embryopathy is an isolated defect. The remaining 30% of the cases coexist with around 300 rare genetic syndromes, such as Van der Woude syndrome, DiGeorge/velocardiofacial syndrome and Treacher Collins syndrome. The annual incidence of this anomaly is estimated at 1/4000 to 1/10,000 births, depending on the ethnic group and geographic location. Cleft lip and alveolus occurs twice more often in boys than in girls, most often on the left side. [5].

Central facial nerve palsy results from damage of the upper motor neurons of the facial nerve, and clinically presents as the paralysis of the lower part of the face. Facial nerve palsy in children can be congenital or acquired. The acquired kind may result from trauma, bacterial and viral infection, neoplasia or be idiopathic. Congenital causes include traumatic delivery and/or genetic syndromes (e.g., Moebius syndrome, Goldenhar syndrome, syringobulbia and Arnold Chiari malformations) [6].

This paper aims to present a patient with multiple congenital anomalies: APW, thumb aplasia, central facial nerve palsy and cleft lip and alveolus. We discuss a possible interrelation between these anomalies in the aspect of tips for diagnostics of rare heart anomalies.

## 2. Case Report

A 40-year-old female patient, who has been a patient of the plastic surgery out-patient clinic from infancy, was born at 37 weeks, after an uneventful pregnancy and delivery (birth weight—2350 g; Apgar score—8). After delivery, she was diagnosed with a right thumb aplasia and right-sided cleft lip and alveolus (Figure 1). The patient’s mother worked (also during pregnancy) in the factory where thermometers with mercury were manufactured, and her family history of congenital malformations was negative (she has a ten years younger healthy sister). None of the anomalies were detected prenatally (due to the lack of routine diagnostics 40 years ago). When the child was 12 months, she was admitted to the plastic surgery clinic for a cleft lip repair, which was performed using the Tennison method. At this stage, preoperatively, a right-sided facial nerve palsy was also diagnosed, involving the lowering of the right corner of the mouth The postoperative course was uneventful (Figure 2). A hand X-ray at the age of 3 years showed four fingers of the right hand with thumb aplasia and a decreased bone age with an incorrect order of wrist bones ossification.

At the age of 3 years, she was hospitalized in the pediatric cardiology department due to a sudden deterioration of her general health. A chest X-ray revealed an increased blood flow through the lungs with symptoms of pulmonary hypertension and an enlarged heart, especially the right ventricle, and a wide mediastinum. A heart catheterization showed APW with a significantly increased pulmonary resistance limiting the left–right shunt through the window, an enlarged left atrium and left ventricle with a preserved systolic function. A physical examination revealed a holosystolic crescendo–decrescendo murmur with the crescendo peak closer to the first heart sound, a high amplitude, split second heart sound over the pulmonary artery (PA) and protosystolic murmur over PA and at the Erb point. The child underwent a total correction of the heart defect in extracorporeal circulation. An aortotomy was performed and the pulmonary artery was cut with a longitudinal APW closure (with a velour patch). One month after the surgery, a protomesosystolic decrescendo murmur was present in the projection of the pulmonary artery valve, apex and Erb point. In an ECHO, a paradoxical movement of the intraventricular septum was observed and the left ventricle was within the upper limit of the norm. The child was discharged from the department and followed-up with regularly. At the age of 17 years, she was hospitalized in the cardiology department for extended examinations. An ECHO examination showed a pulmonary valve regurgitation (I/II degree), PA with a widened trunk, widened ascending aorta with narrowing at the isthmus level and acceleration of flow, but without features of coarctation. The exercise tolerance test was correct, without signs of hypertrophy and damage to the heart muscle in resting ECG. A twenty-four-hour Holter monitoring also did not show any disturbances.

The patient was again hospitalized in the plastic surgery clinic when she was 12 years old and a bone graft from the iliac crest to the alveolus was performed. Then, she underwent the correction of a post-cleft nose deformity (auricular cartilage graft to the nasal ala, septoplasty, a nasal tip plasty) (Figure 2).

Due to chronic headaches, in 2018, the patient had a head MRI performed, which showed a colloidal pineal cyst (8 × 6 × 5 mm), currently with no indications for treatment.

A long-term follow-up showed normal social functioning and the patient refused any intervention involving the hand (i.e., index pollicization), correction of subclinical facial nerve palsy and alveolus and nose defect (Figure 3 and Figure 4). The patient’s chest X-ray reveled dilated bronchial vessels, and an ECHO examination showed an increased peak velocity of blood in the PA, indicating stenosis (at the level of the APW repair), which was stable and subclinical (Figure 5). The patient had a healthy female child without any congenital anomalies.

## 3. Discussion

In this case report, we presented a patient with multiple congenital anomalies: aortopulmonary window, thumb aplasia, central facial nerve palsy and cleft lip and alveolus. To our knowledge, this is the first report of such anomalies coexisting. However, if similar cases are reported, it should be verified whether the observed defects are just a unique coincidence or if they constitute a very rare but repeatable syndrome.

The presence of certain defects (hand anomalies, cleft lip and alveolus (primary palate) and congenital facial nerve palsy), and, in particular, such a rare defect as thumb aplasia, may indicate the need for a careful fetal ultrasound extended by an ECHO examination performed in a prenatal cardiology center. Such a detailed examination could reveal rare defects, e.g., APW, which may not be detected by an unexperienced examiner. It could contribute to a more accurate prenatal diagnosis and planning of postnatal care, including surgical treatment. Such a treatment is indicated as early as possible to prevent congestive heart failure and pulmonary hypertension. Elliotson, in 1830, described APW for the first time [7], while Collinet et al., in 2002, determined the first prenatal echocardiographic diagnosis [8]. The detection of APW in fetal echocardiography is, however, challenging, as it involves the need to visualize the aortopulmonary septum [9]. The surgical closure should be performed as soon as possible—without operation, 40–50% of patients will die during the first year of life [10]. An early operation can prevent congestive heart failure caused by a high pulmonary blood flow [9]. APW may be a part of a very rare Berry syndrome characterized by a constellation of heart abnormalities: aortopulmonary window (APW), aortic origin of the right pulmonary artery (AORPA), an interrupted aortic arch (IAA) or hypoplastic aortic arch (HAA) or coarctation of the aorta (CoA) and intact ventricular septum [11,12]. This rare combination of cardiac anomalies can be treated surgically, but also, in this case, an early diagnosis is critical to avoid pulmonary hypertension. However, APW has not yet been described as a part of any “noncardiac” syndrome, but it was reported in a coincidence with Axenfeld–Rieger syndrome. The syndrome is a genetic disorder, in which the major physical condition is an anterior segment dysgenesis of the eye, but other non-ocular congenital anomalies may be present: craniofacial dysmorphism (hypertelorism, telecanthus, maxillary hypoplasia with flattening of the mid-face, a prominent forehead and a broad, flat nasal bridge), dental anomalies (microdontia or hypodontia), redundant periumbilical skin and urogenital abnormalities [13].

As supported by the literature, fetuses with congenital heart malformations (particularly in cyanotic) are often small for their gestational age (SGA) at birth and their head circumference is small. This was also observed in our patient who was born with a low birth weight (data on head circumference were not available), but the SGA was not detected prenatally due to limited diagnostic techniques 40 years ago [14]. Although multiple malformations increased the risk of preterm birth and preeclampsia, the course of the pregnancy in this case was uneventful [15].

Facial nerve palsy in the reported patient may be considered subclinical as it involved only a slight asymmetry of lips’ angle. A clinical picture and adverse history for any birth trauma may have suggested a form of congenital unilateral lower lip palsy (CULLP). This condition is often described in Cayler syndrome involving cardiac defects caused by the 22q11.2 deletion syndrome [16]. However, in the case of our patient, no agenesis or hypoplasia of the depressor anguli oris muscle was detected, and APW is not a typical defect in this syndrome. Moreover, in the case of cardiofacial syndrome, facial asymmetry is present when crying, but not at rest—as in our case. Moebius syndrome, which could be another cause of congenital facial nerve palsy, is bilateral and also involves abducens nerve palsy, which was not present in our patient [17]. Congenital facial palsy (CFP) seems the most appropriate diagnosis in the reported patient. Although the most common causes (perinatal trauma and intrapartum compression) were not confirmed, other possible causes, such as the congenital aplasia of the nucleus, were not excluded [18].

Another syndrome that could be considered in the presented patient is oral–facial–digital syndrome (OFDS). It includes a heterogeneous group of disorders that affect the oral cavity, face and digits. This syndrome has fourteen subtypes, of which two are unclassified OFDSs with overlapping phenotypes. The majority of OFDSs are inherited as an autosomal recessive syndrome, except OFD I and VIII (X-linked). Additionally, the syndrome may involve the brain, kidneys, pancreas, liver, skin and bones. Cardiac defects, but not APW, are observed in types II, VI, IX, XII and XIII, and in the unclassified two. Although cleft lip is a common malformation in OFDS, thumb atresia is not a component of any type of OFDS. In this syndrome, hand anomalies usually involve the brachydactyly, polydactyly, clinodactyly and syndactyly, with a common involvement of feet [19].

Although we analyzed multiple phenotypes and clinical manifestations included in a base OMIM^®^—Online Mendelian Inheritance in Man^®^—, we did not find a genetic syndrome that matched the presented patient or involved at least two of the observed anomalies. This could suggest the role of environmental factors in the prenatal development and the random coexistence of the observed anomalies. Moreover, all anomalies must have occurred in similar time frames of embryological development: heart septation—4–7th week (specifically 8th week); primary palate fusion—4–7th week; thumb development—5–8th week; cranial nerves—from 4th week. To verify this, we also considered a possible exposure to mercury (mother’s occupational risk) in the aspect of the patient’s clinical picture. The Spanish study conducted by Murcia et al. revealed that a reduced placental and fetal growth may be associated with mercury exposure from seafood consumption during prenatal life [20]. Vigeh et al. found that higher blood mercury levels during the first and second trimesters of gestation led to a lower birth weight. In this study, mercury levels in the umbilical cord blood were twice as high as the maternal blood levels [21]. In addition, a high-level prenatal exposure to methylmercury (fetal-type Minamata disease) leads to fetal brain damage and subsequent neurological symptoms, but does not cause congenital malformations. Children exposed to mercury prenatally present disturbances in mental development in later life and central neurological disorders (problems with walking, sitting, hand tremor and postural sway) [22]. Such symptoms were not present in our patient, as her mental and motor development was undisturbed with normal social functioning. However, most studies concerning fetal mercury exposure report the results of the consumption of contaminated food, which is known to have a neurotoxic effect to the infants exposed in utero. The patient’s mother was exposed to liquid mercury previously used in thermometer and sphygmomanometer manufacture. Liquid mercury vaporizes and stays in the air, so her exposure seemed to be significant. Nothing is known about the safety measures and sanitary conditions in the place of her occupation. In such work, the major route of exposure was via the inhalation of mercury vapor, but it can also be absorbed cutaneously [23]. Mercury in maternal blood is able to cross the placenta, reach the fetal circulation and cause neurotoxicity, nephrotoxicity, teratogenicity, cardiovascular disease and possible carcinogenicity [23]. A meta-analysis from China showed that occupational exposure to mercury could cause stillbirth, a low birth weight and birth defects [24]. However, other authors did not find any increased rates of miscarriage or birth defects in occupationally exposed women [25], so finding a direct association between the anomalies observed in our patient and her mother’s occupational risk seems to be questionable.

The main limitation of the reported case was a lack of genetic examinations, as such techniques were not available when the patient was diagnosed during childhood. However, there is no specific genetic factor known to cause such malformations (known syndromes were excluded based on the clinical picture). The medical history did not reveal any congenital malformations in her family and the patient had a healthy child, so she was not interested in undergoing genetic tests. Further, it is impossible to document whether her mother truly had mercury poisoning during her pregnancy.

To sum up, this case report showed that the presence of certain defects (thumb aplasia) may indicate the need for a careful fetal examination extended by a fetal ECHO performed in a reference center of prenatal cardiology. The coexistence or syndromic character of the presented malformations should be verified in the future if more such cases are described.

## Figures and Tables

**Figure 1 diagnostics-12-00569-f001:**
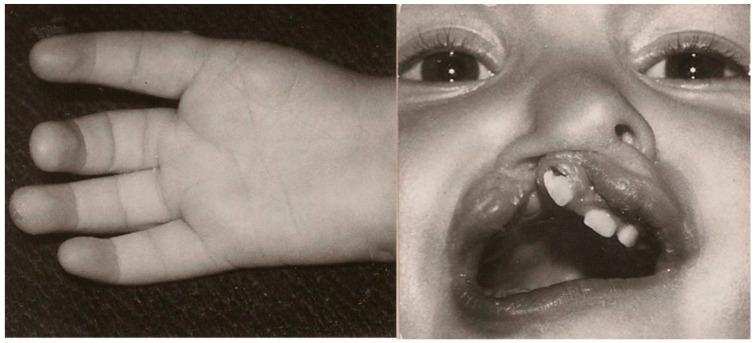
Right hand with thumb aplasia (palmar view) and cleft lip and alveolus at 12 months.

**Figure 2 diagnostics-12-00569-f002:**
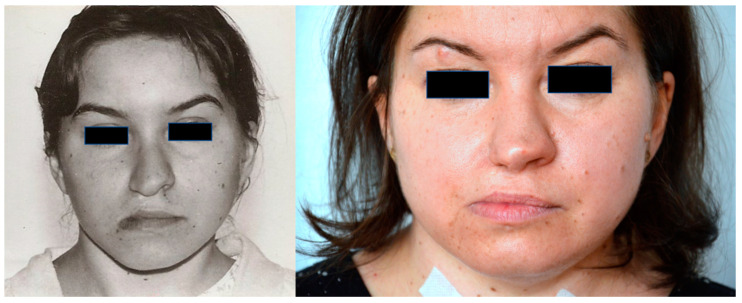
Post cleft repair, at the age of 14 (**on the left**), and at the age of 40 (**on the right**), lowering of the right mouth angle.

**Figure 3 diagnostics-12-00569-f003:**
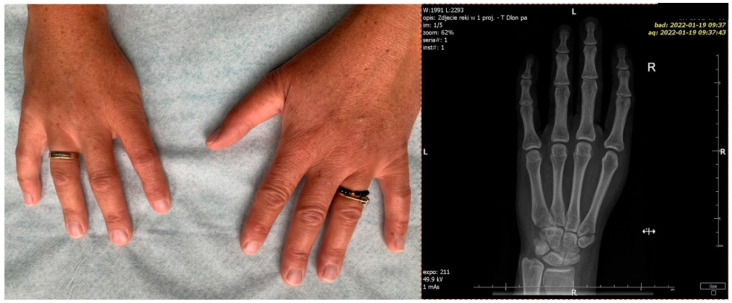
Patient’s hands at the age of 40. Right hand X-ray: absence of I metacarpal and thumb phalanges, hypoplasia of the scaphoid bone and styloid process.

**Figure 4 diagnostics-12-00569-f004:**
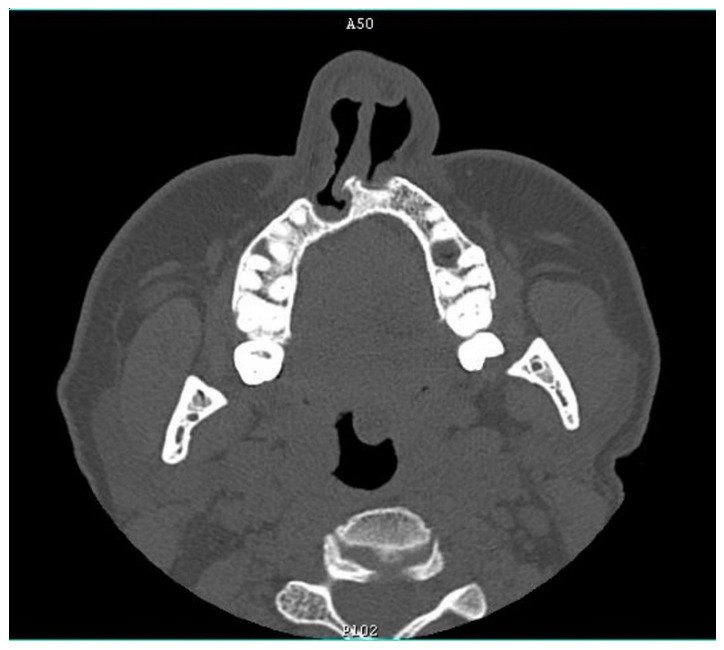
Defect of alveolus and nose at the age of 40 (CT scan).

**Figure 5 diagnostics-12-00569-f005:**
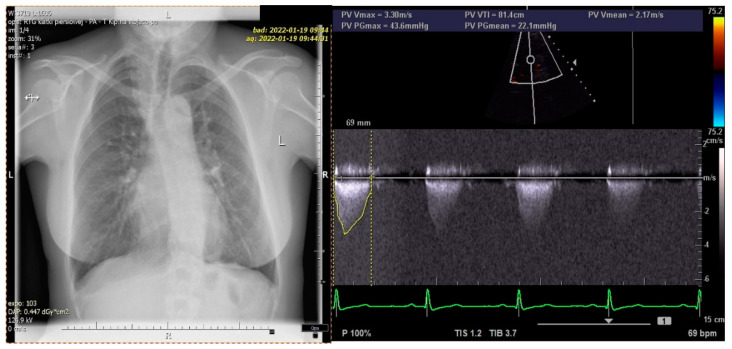
Chest X-ray and ECHO at the age of 40: dilated bronchial vessels (**left**), the peak velocity of blood in PA (at the level of the repaired segment) = 3.3 m/s (**right**).

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
