# Peer review of "Coexistence of Thumb Aplasia and Cleft Lip and Alveolus with Aortopulmonary Window—A Tip for Prenatal Diagnostics for Rare Heart Anomalies"

_diagnostics, 2022, doi:10.3390/diagnostics12030569_

Round 1

Reviewer 1 Report

Kasielska-Trojan A. et Al. present an original clinical case of a female patient with post-natal diagnostic of multiple congenital anomalies - aortopulmonary window (APW), right thumb aplasia, facial nerve palsy and cleft lip and alveolus. Their association is quite unique and leads to an interesting discussion concerning the etiopathogenesis (potentially genetic syndrome + environment factors (mercury poisoning during her mother’s pregnancy)).

The clinical case is very well documented, including high quality pictures and has a 40-year follow-up. Also, the clinical outcome is positive, as the patient herself managed to have a normal life following corrective surgeries – including APW successful correction for symptomatic right heart failure due to left-to-right shunt PAH, with a clinically non-significant residual pulmonary stenosis.

The article is written in good English, easy to follow and very well documented reference-wise.

The only limitation is the lack genetical testing and the impossibility of documenting whether her mother truly has had mercury poisoning during pregnancy.

I recommend the article for publication.

Minor comments:

  • Abstract – page 1 line 11: change « cause » to « causes »
  • Page 7 line 231 : change « reports » to « report »

Author Response

Dear Editor and Reviewers,

Thank you for your interest in our case report entitled " Coexistence of thumb aplasia and cleft lip and alveolus with aortopulmonary window – a tip for prenatal diagnostics for rare heart anomalies”. We would like to thank for the valuable comments. In this revision we addressed all your comments. We hope that this revision meets with your approval.

Sincerely,

The Authors

Responses to Reviewers’ comments:

Reviewer 1

I recommend the article for publication.

We would like to thank the Reviewer for the approval!

Minor comments:

Abstract – page 1 line 11: change « cause » to « causes »

Page 7 line 231 : change « reports » to « report »

We also added a sentence highlighting limitation of the report:

“However, it is impossible to document whether her mother truly has had mercury poisoning during pregnancy.”

All done. Thank you for detailed proofreading.

Reviewer 2

-Line 46. Insert full stop

-capitals - line 61.

Done.

-In general, the manuscript may be proof-read by a native speaker

The paper was proof-read by our university  lecturer.

Apart from these minor reports: phenomenal manuscript. The authors are to be applauded for the extensive (40y!) follow-up and the meticulous preparation of the figures

We would like to thank the Reviewer for the appreciation!

Reviewer 2 Report

Interesting case:

-Line 46. Insert full stop

-capitals - line 61.

-In general, the manuscript may be proof-read by a native speaker

Apart from these minor reports: phenomenal manuscript. The authors are to be applauded for the extensive (40y!) follow-up and the meticulous preparation of the figures

Author Response

Dear Editor and Reviewers,

Thank you for your interest in our case report entitled " Coexistence of thumb aplasia and cleft lip and alveolus with aortopulmonary window – a tip for prenatal diagnostics for rare heart anomalies”. We would like to thank for the valuable comments. In this revision we addressed all your comments. We hope that this revision meets with your approval.

Sincerely,

The Authors

Responses to Reviewers’ comments:

Reviewer 1

I recommend the article for publication.

We would like to thank the Reviewer for the approval!

Minor comments:

Abstract – page 1 line 11: change « cause » to « causes »

Page 7 line 231 : change « reports » to « report »

We also added a sentence highlighting limitation of the report (as suggested):

“However, it is impossible to document whether her mother truly has had mercury poisoning during pregnancy.”

All done. Thank you for detailed proofreading.

Reviewer 2

-Line 46. Insert full stop

-capitals - line 61.

Done.

-In general, the manuscript may be proof-read by a native speaker

The paper was proof-read by our university lecturer

Apart from these minor reports: phenomenal manuscript. The authors are to be applauded for the extensive (40y!) follow-up and the meticulous preparation of the figures

We would like to thank the Reviewer for the appreciation!